# A Tail of Eternal Inflation

Timothy Cohen[1], Daniel Green[2], and Akhil Premkumar[2]

[1]*Institute for Fundamental Science, University of Oregon, Eugene, OR 97403, USA*

[2]*Department of Physics, University of California at San Diego, La Jolla, CA 92093, USA*

## Abstract

Non-trivial inflaton self-interactions can yield calculable signatures of primordial non-Gaussianity that are measurable in cosmic surveys. Surprisingly, we find that the phase transition to slow-roll eternal inflation is often incalculable in the same models. Instead, this transition is sensitive to the non-Gaussian tail of the distribution of scalar fluctuations, which probes physics inside the horizon, potentially beyond the cutoff scale of the Effective Field Theory of Inflation. We demonstrate this fact directly by calculating non-Gaussian corrections to Stochastic Inflation within the framework of Soft de Sitter Effective Theory, from which we derive the associated probability distribution for the scalar fluctuations. We find parameter space consistent with current observations and weak coupling at horizon crossing in which the large fluctuations relevant for eternal inflation can only be determined by appealing to a UV completion. We also show this breakdown of the perturbative description is required for the de Sitter entropy to reflect the number of de Sitter microstates.

# 1 Introduction

Developing a complete picture of physics in de Sitter space remains one of the great unsolved problems in theoretical physics [1–3]. The issues appear in many guises. On the practical side, we do not have a rigorous (non-perturbative) definition of cosmological observables [1,4]. More conceptually, confusions abound when attempting to characterize the eternal inflating phase [5–7]. Meanwhile, these significant challenges do not seem to impede our ability to make quantitative predictions for the universe we inhabit. Weak coupling allows us to calculate and understand the structure of observable correlation functions as a controlled approximation. Yet, our goal in this paper is to demonstrate, for the first time, that there are fundamental questions about our own patch of the universe whose answers are not calculable in perturbation theory, *e.g.* the possibility that our universe is eternally inflating.

Cosmological observations suggest that the large scale structures in our universe were seeded during inflation, a period of quasi-de Sitter expansion [8–10]. The observable implications of inflation can be captured by an Effective Field Theory (EFT) framework [11,12]. Much progress has been made in understanding how to calculate the statistical predictions of inflation perturbatively [13, 14]. A notable recent advance is the cosmological bootstrap, which aims to reconstruct inflationary observables directly from locality and causality [15–21]. Much of the interest in the structure of cosmological correlators centers on the possible signatures of primordial non-Gaussianity, since this provides an observational window into the particle content and interactions that played a role during inflation [22].

The fact that the observational and conceptual aspects of cosmology are decoupled is a simple consequence of dimensional analysis, which additionally underlies the validity of the EFT of Inflation approach. There is a significant separation between the two energy scales $H$ and $f_\pi$ that characterize inflation [23] (see also [12, 24–35]), as illustrated in Fig. 1. The EFT of Inflation can be framed in terms of the spontaneous breaking of time translation symmetry, where the associated Goldstone boson $\pi$ describes the scalar density fluctuations. The universal scale describing the dynamics of the fluctuations is $f_\pi^2 \simeq |\dot\phi|$, where $\dot\phi \neq 0$ is the order parameter for the breaking of time translation invariance, and $\phi$ is a fundamental scalar in most concrete UV models. Typical de Sitter fluctuations are produced with a characteristic energy set by the Hubble parameter during inflation $H$, which results in there being a de Sitter temperature $2\pi T_{\mathrm{dS}} = H$. In more detail, an emergent scalar degree of freedom $\zeta$ experiences adiabatic fluctuations, whose amplitude is $2\pi^2 A_s \equiv \Delta_\zeta = H^4/(2f_\pi^4)$. In our patch of the universe, measurements of the cosmic microwave background imply $A_s = 2.1 \simeq 10^{-9}$ [36], and so we can infer $f_\pi \simeq 59H$. This tells us that $f_\pi \gg H$ is a good approximation in our universe. As we explore the physics of de Sitter space and the relation to eternal inflation in this work, we will also consider $f_\pi$ and $H$ to be free parameters, for example the parameter space where $f_\pi \simeq H$.

Primordial non-Gaussianity in single-field inflation arises through derivative interac-

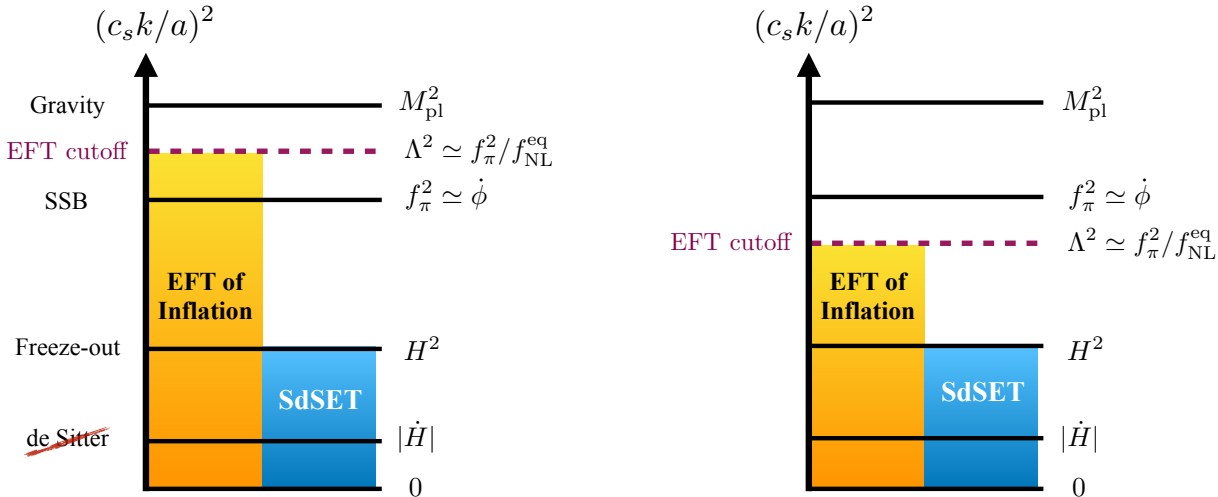

Figure 1: The relevant energy scales for single field inflation and the regimes of validity of the EFT of Inflation [orange] and Soft de Sitter Effective Theory (SdSET) [blue]. The background time evolution leads to a scale $f_\pi$, below which the time translation symmetry is spontaneously broken (SSB). Primordial non-Gaussianity arises from interactions that are suppressed by the EFT cutoff scale $\Lambda$, which is related to the amplitude for equilateral non-Gaussianity $f_{NL}^{eq}$ by $\Lambda^2 \simeq f_\pi^2/f_{NL}^{eq}$. Requiring the EFT of Inflation is weakly coupled at horizon crossing allows both $\Lambda > f_\pi$ [left] and $\Lambda < f_\pi$ [right]. Deviations from a de Sitter background arise at $|\dot{H}| \ll H^2$.

tions that are suppressed by some dimensionful UV scale $\Lambda$.[1] While the precise relationship to the amplitude of equilateral non-Gaussianity $f_{NL}^{eq}$ varies among different possible models, the scaling relation $f_{NL}^{eq} \simeq f_\pi^2/\Lambda^2$ is universal. Given the current constraints from Planck, $f_{NL}^{eq} = -26 \pm 47$ (68% confidence interval) [42], the region of parameter space where $\Lambda^2 \ll f_\pi^2$ remains a viable possibility. On the other hand, canonical models of slow roll inflation require that $f_{NL}^{eq} < 1$ so that the background evolution $\dot{\phi}$ is calculable in the weakly coupled regime [31,32,43]. Nevertheless, a number of compelling models such as DBI inflation [44], models that utilizes non-trivial field space curvature [45,46], and those involving interactions with massive fields [47,48] can easily produce $f_{NL}^{eq} \gg 1$ self-consistently. It is only essential that $\Lambda > H$ in order to reliably calculate the observational predictions using perturbation theory [23]. In this work, we revisit whether $\Lambda > H$ is sufficient ensure perturbative control over all quantities of interest.

For cosmological correlators, all of the thorny issues of observables in de Sitter are under control as long as one is in the perturbative regime where $H^2/M_{pl}^2$, $H^2/f_\pi^2$, and $H^2/\Lambda^2$ are all small. To an excellent approximation, inflation is described by a fixed background geometry in which the scalar fluctuations evolve. In the absence of non-Gaussianity, even the onset of slow-roll eternal inflation is calculable and arises when

---

[1] Here we are assuming scale invariant non-Gaussianity. Scale dependent signals [37], such as models of resonant non-Gaussianity [38], are also possible. As these model also leave signatures in the power spectrum [39–41], we will not consider them further to ensure a clean separation between the Gaussian and non-Gaussian effects.

$\Delta_\zeta \geq \pi^2/3$ [49]. Building on this, one might expect then that the phase transition to eternal inflation in models with primordial non-Gaussianity can also be calculated as a perturbative expansion in $H^2/\Lambda^2$. Remarkably, this intuition is wrong. We will show that there are corrections to the expansion that scale as $f_\pi^2/\Lambda^2$. This implies that when $\Lambda \ll f_\pi$, there should be incalculable corrections within the EFT of Inflation [11, 12], even when all the $N$-point correlators are calculable in perturbation theory. Our goal is demonstrate this result and explain why it occurs.

At a qualitative level, the onset of slow-roll eternal inflation occurs when the amplitude of quantum fluctuations exceeds that classical motion of the field [5, 6]. In canonical slow-roll inflation $f_\pi^2 = \dot\phi$, so that the classical distance moved in a Hubble time ($\Delta t = H^{-1}$) is $(\Delta\phi)_{\text{classical}} \simeq f_\pi^2/H$. Meanwhile, the Gaussian *quantum* fluctuation introduce an effective noise in the motion of the field with an amplitude set by the expansion rate, $(\Delta\phi)_{\text{noise}} \simeq H$. Slow-roll eternal inflation occurs when these two types of field excursions are of the same order, $(\Delta\phi)_{\text{classical}} \simeq (\Delta\phi)_{\text{noise}}$, which happens when $H^2 \simeq f_\pi^2$ or equivalently when $\Delta_\zeta \simeq 1$. However, implicit to this argument is that the rate for generating fluctuations that are larger than $H$ is negligible. If instead there was a non-negligible rate for quantum fluctuations from the tail of the distribution such that $(\Delta\phi)_{\text{tail}} \simeq f_\pi^2/H$ with $f_\pi > H$, these larger quantum fluctuations could be the dominant effect that would determine the onset of eternal inflation. The probability of such a large fluctuation is exponentially small for Gaussian theories. However, primordial non-Gaussianity could, in principle, increase the rate of these large fluctuations such that they dominate the onset of eternal inflation. Noting that the energy scale associated with such non-Gaussian quantum fluctuations is $(\dot\phi)_{\text{tail}} \simeq H(\Delta\phi)_{\text{tail}} \simeq f_\pi^2$, these fluctuations would correspond to physics above the UV cutoff for models with $\Lambda < f_\pi$.

In this paper, we use Soft de Sitter Effective Theory[2] (SdSET) [50, 51] to calculate corrections to Stochastic Inflation [52] (see also [53–65]), which allows us to demonstrate that the onset of eternal inflation is incalculable when $\Lambda < f_\pi$. This occurs because large field variations (corresponding to the tail of the probability distribution) are probes of high energy physics during inflation. When $\Lambda < f_\pi$, the onset of eternal inflation is sensitive to the regime where the EFT does not apply. Concretely, the blue shaded region in Fig. 2 naively corresponds to eternal inflation in our universe and signals this breakdown. Interestingly, this parameter space overlaps the regions allowed by current observations and weak coupling at horizon crossing. The source of the issue is that correctly modeling the tails of the probability distributions requires a non-perturbative calculation of the transition probabilities that go beyond the perturbative contributions that are included in the Stochastic Inflation framework. We interpret the blue region as providing a sharp

---

[2]While the EFT of Inflation and SdSET have overlapping regions of validity, SdSET makes manifest the long wavelength behavior of the fluctuations in the universe, particularly with regards to IR divergences and their resummation via the dynamical renormalization group (RG). This property of SdSET is essential for deriving the results in this paper.

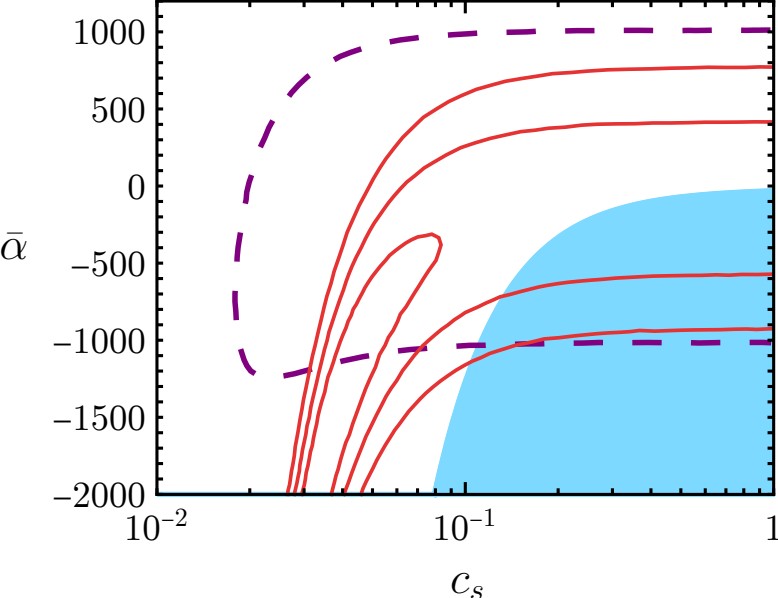

Figure 2: The solid blue region shows where primordial non-Gaussianity naively implies eternal inflation, suggesting a breakdown of Stochastic Inflation and/or the EFT of Inflation. The region allowed by perturbative unitarity at horizon crossing, derived in [35], corresponds to the region enclosed by the dashed purple line. The current 1-,2- and 3-$\sigma$ limits from Planck [42] are shown as solid red lines. We see there is a significant region where the calculation of the transition to external inflation is breaking down, that is nonetheless consistent with the theory being weakly coupled at horizon crossing (as determined by unitarity) and current observations. The parameters $\bar{\alpha}$ and $c_s$ are related to the two allowed cubic couplings in the EFT of Inflation, as defined in Sec. 3.

bound, akin to a perturbative unitarity bound at the energy scale $f_\pi$. Otherwise, as we show below, the EFT predictions would be inconsistent with interpreting the de Sitter entropy [66] as resulting from a finite number of degrees of freedom (see *e.g.* [2,3,67–71]).

This work is adds a novel direction to the vast literature on the perturbative regime inflationary fluctuations [23–35] and the implications for eternal inflation [72–77]. Prior discussions of eternal inflation are relevant for the parameter space with $f_\pi \simeq H$ ($\Delta_\zeta = \mathcal{O}(1)$), as this is the only regime of canonical slow-roll inflation where eternal inflation can occur. In any parameter regime, it is a necessary condition that the theory is weakly coupled at horizon crossing, $\Lambda > H$, for calculations to be under control. In the context of previous discussions of eternal inflation where $f_\pi \simeq H$, the breakdown of weak coupling at horizon crossing is indistinguishable from the breakdown of the Stochastic framework. In contrast, most of the discussion in our paper applies to our own observable universe where $f_\pi \simeq 59H$ ($\Delta_\zeta \simeq 4.1 \times 10^{-8}$), and where the theories of inflation of interest are weakly coupled at horizon crossing. One would not expect eternal inflation in this regime. However, although much is known about the structure of Stochastic Inflation in canonical

slow roll models [52–65], before this work it was not known how to include the non-Gaussian corrections into the Stochastic Inflation framework. These are exactly the new ingredients that are required to ask questions about the phase transition to eternal inflation. Our concrete results will show that there is a breakdown in the calculation that is signaled by the apparent onset of eternal inflation in the regime $f_\pi \gg H$. This failure of Stochastic Inflation is only relevant when attempting to predict the tail of the distribution of scalar fluctuations and is distinct from having control over perturbative calculations at horizon crossing.

The paper is organized as follows. In Sec. 2, we calculate the first higher derivative correction to Stochastic Inflation from primordial non-Gaussianity in Single-Field Inflation. In Sec. 3, we solve these corrected equations and use the results to compute the onset of eternal inflation. We apply these results to interpretation of the de Sitter entropy in Sec. 4. In Sec. 5, we interpret the surprising dependence on non-Gaussian fluctuation as a breakdown of Stochastic Inflation requiring a UV calculation of the underlying transition amplitudes. We conclude in Sec. 6. Two appendices give background for these results. In App. A, we review aspects of single field inflation that are essential for understanding the key results in this paper. In App. B, we provide an alternate derivation of our solution to the corrected Fokker-Planck equation using the Fourier transform and the method of steepest descents.

## 2   Non-Gaussian Corrections to Stochastic Inflation

Light scalar fields in quasi de Sitter space, such as the inflaton, undergo random quantum fluctuations. In perturbation theory, these fluctuations give rise to large infrared (IR) effects, which can be resummed using the framework known as Stochastic Inflation [52, 53, 56]. This gives rise to a Fokker-Planck equation that determines the evolution of the probability distribution for the local value of the field.

The canonical formation of Stochastic Inflation provides a leading order prediction for the field's evolution. Interactions correct the Fokker-Planck equation at higher orders, which can be represented on general grounds as [51] (see also [78, 79])

$$\frac{\partial}{\partial t}P(\phi,t) = \sum_{n=2}^{\infty} \frac{1}{n!} \frac{\partial^n}{\partial \phi^n} \left[ \sum_{m=0}^{\infty} \frac{1}{m!} \Omega_n^{(m)} \phi^m P(\phi,t) \right] + \frac{1}{3H} \frac{\partial}{\partial \phi} \left[ V'(\phi) P(\phi,t) \right] . \qquad (2.1)$$

The term proportional to $V'(\phi)$ is just the classical evolution of the field, and the rest of the terms account for the quantum fluctuations. The original formulation due to Starobinski applies to leading order in the coupling,[3] in which case the quantum noise is given by $\Omega_2^{(m=0)} = H^3/(8\pi^2)$ with all other $\Omega_n^{(m)} = 0$.

---

[3]See *e.g.* [51] for a derivation of the power counting for Stochastic Inflation.

Given the intuitive description of Stochastic Inflation, it might seem surprising that calculating these higher order corrections remained elusive until recently [51, 78, 79]. It had often been suggested that the Stochastic framework is related to IR divergences in dS [50, 78–88]. Leveraging this insight to systematically improve the framework naturally results in the SdSET approach [50]. The SdSET converts the full theory IR divergences into EFT UV divergences in the usual sense (see *e.g.* [89]). This allows one to resum full theory IR divergences using the usual RG playbook within the EFT. Specifically, Stochastic Inflation is equivalent to the (dynamical) RG for SdSET composite operators. The contributions from the quantum noise can be extracted from operator mixing under time evolution, which takes the generic form

$$\frac{\partial}{\partial \mathtt{t}} \left\langle \varphi_+^N \right\rangle = \sum_{m=0}^{\infty} \sum_{n=1}^{N+m} \tilde{\Omega}_n^{(m)} \, (-1)^n \binom{N}{n} \left\langle \varphi_+^{N-n+m} \right\rangle \; , \tag{2.2}$$

where for a massless scalar field $\phi$, we identify $\varphi_+$ as the growing mode mode such that $\phi \to H\varphi_+$. We also defined $\mathtt{t} = Ht$ and $\tilde{\Omega}_n^{(m)} = H^{m-n-1}\Omega_n^{(m)}$ to simplify the expression in terms of $\varphi_+$. Finally, $V'(\phi)$ is replaced by $\tilde{\Omega}_1^{(m)}$, which also receives corrections at higher orders.

## 2.1   Stochastic Inflation for Single Field Inflation

The first higher-derivative correction to this framework was calculated in [51] assuming the UV model was $\lambda\phi^4$ in fixed dS. We will now extend these results to single-field inflation. This is a non-trivial generalization both because the metric fluctuates (the background is no longer fixed dS), and these fluctuations are subject additional constraints from the diffeomorphism invariance. The corrections to Stochastic Inflation are most transparent when expressed in terms of the scalar metric fluctuation $\zeta$. This choice is particularly useful because $\zeta$ transforms non-linearly under large diffeomorphisms [90–93]

$$D_{\mathrm{NL}} : \delta\zeta = -1 - \vec{x} \cdot \vec{\partial}_{\vec{x}}\zeta \tag{2.3a}$$

$$K_{\mathrm{NL}}^i : \delta\zeta = -2x^i - 2x^i \left( \vec{x} \cdot \vec{\partial}_{\vec{x}}\zeta \right) + x^2 \partial^i \zeta \; . \tag{2.3b}$$

The Ward identities associated with these symmetries [94,95] impose constraints on correlation functions that are also known as the single field consistency conditions [90,91]. The above transformation uniquely fixes the definition of $\zeta$, and it ensures our results will be free from field redefinition ambiguities and scheme dependence [96]. The important implication for our purposes here is that these non-linearly realized symmetries fix the form of possible corrections to the Stochastic Inflation framework. This is already known for the properties of $\zeta(\vec{x})$ at separated points, where it leads to the all-orders conservation of $\zeta(\vec{k})$, namely $\dot{\zeta}(\vec{k}) \to 0$ as an operator statement in the limit $k/(aH) \to 0$ [50, 55, 97, 98]. From

Eq. (2.2), we see that applying these symmetries to Stochastic Inflation is the same as extending the operator statements to products of $\zeta$'s at coincident points, *i.e.*, composite operators built from $\zeta$.

By power counting in the SdSET, the dynamical RG of any light field is necessarily ultra-local in space, in that it contains no derivatives. This implies that the most general possible result must take the form

$$\frac{\partial}{\partial t}\zeta^N(\vec{x}, t) = \sum_M \Gamma_M^N(t)\zeta^M(\vec{x}, t) \, . \tag{2.4}$$

Applying $D_{\mathrm{NL}}$ from Eq. (2.3a) to the both sides of the equation implies

$$\frac{\partial}{\partial t}N\zeta^{N-1}(\vec{x}, t) = \sum_M M\Gamma_M^N(t)\zeta^{M-1}(\vec{x}, t) \, . \tag{2.5}$$

Substituting Eq. (2.4) on the left-hand side yields

$$N\sum_M \Gamma_M^{N-1}(t)\zeta^M(\vec{x}, t) = \sum_M M\Gamma_M^N(t)\zeta^{M-1}(\vec{x}, t) \, . \tag{2.6}$$

Matching the powers of $\zeta$, we find

$$(N+1)\Gamma_M^N = (M+1)\Gamma_{M+1}^{N+1} \, . \tag{2.7}$$

We demand $\Gamma_M^N = 0$ if $N < 0$ or $M < 0$, since operators with fields in the denominator are unphysical. If we assume $N > 0$ and the existence of a first non-zero anomalous dimension $\Gamma_0^n \equiv \gamma_n \neq 0$ for some $n$, the solution to Eq. (2.7) becomes

$$\Gamma_M^N = \gamma_n \delta_{M,N-n} \prod_{\ell=n+1}^N \frac{\ell}{\ell-n} = \binom{N}{n}\gamma_n \delta_{M,N-n} \, . \tag{2.8}$$

Summing over all possible $\gamma_n$, we have

$$\frac{\partial}{\partial t}\zeta^N(\vec{x}, t) = \sum_n \gamma_n \binom{N}{n}\zeta^{N-n}(\vec{x}, t) \, . \tag{2.9}$$

Finally, we apply the relation between the operator mixing language and Stochastic Inflation (see *e.g.* [51]), which leads to the following general form for the time evolution of the probability distribution of $\zeta$:

$$\frac{\partial}{\partial t}P(\zeta, t) = \sum_{n\geq 2}(-1)^n\frac{\gamma_n}{n!}\frac{\partial^n}{\partial\zeta^n}P(\zeta, t) \, . \tag{2.10}$$

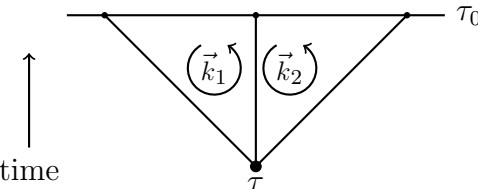

Figure 3: Illustration of the correlation function $\langle\zeta^3(\vec{x}=0)\rangle$ represented as an integral over the bispectrum $B(k_1, k_2, k_3)$ computed at tree-level. The momentum integration is analogous to a two-loop integral.

This result makes intuitive sense: in order to preserve the nonlinear symmetry, the generalization of the Fokker-Planck equation can only depend on derivatives of $\zeta$ (no explicit factors of $\zeta$ appear).

We see from the above result, that we can calculate all corrections to Stochastic Inflation from the mixing coefficients $\zeta^n \to \mathbb{1}$, where $\mathbb{1}$ is the identity operator. For $n = 2$, this is the usual Gaussian (quantum) noise contribution to Stochastic Inflation such that

$$\int \frac{\mathrm{d}^3 k}{(2\pi)^3} \langle \zeta(\vec{k})\zeta(\vec{k}')\rangle = 2\gamma_2 \log aH/K \quad \to \quad \gamma_2 = \frac{\Delta_\zeta}{4\pi^2} \ , \tag{2.11}$$

where $K$ is an IR regulator and $\Delta_\zeta$ is the amplitude of the power spectrum,[4]

$$\langle \zeta(\vec{k})\zeta(\vec{k}')\rangle = \Delta_\zeta k^{-3+(n_s-1)}(2\pi)^3\delta(\vec{k}+\vec{k}') \ . \tag{2.12}$$

Previous studies of stochastic effects in single-field inflation were limited to this contribution and the classical drift from the potential.

We are interested in computing the leading non-Gaussian contribution, which starts at $n = 3$. As we are simply calculating the mixing of operators under dynamical RG, the coefficient of the $n = 3$ term is determined by the logarithmic divergence in the two point function of $\zeta^3$ and $\mathbb{1}$, $i.e.$, the one-point function of $\zeta^3$. This can be calculated, as illustrated in Fig. 3, from the bispectrum (three-point function) via

$$\langle\zeta^3(\vec{x}=0)\rangle = \int \frac{\mathrm{d}^3 k_1 \mathrm{d}^3 k_2 \mathrm{d}^3 k_3}{(2\pi)^3} \langle \zeta(\vec{k}_1)\zeta(\vec{k}_2)\zeta(\vec{k}_3)\rangle \ . \tag{2.13}$$

In single-field inflation, there are two contributions to this three-point function arising

---

[4]Here we are defining $\Delta_\zeta$ so that $\Delta_\zeta = 2\pi^2 A_s$ [99].

from the $\dot{\zeta}\partial_i\zeta\partial^i\zeta$ and $\dot{\zeta}^3$ interactions, which are given by [100]

$$B_{\dot{\zeta}(\partial_i\zeta)^2}(k_1, k_2, k_3) = -\frac{1}{4}\left(1 - \frac{1}{c_s^2}\right)\Delta_\zeta^2$$

$$\times \frac{(24K_3{}^6 - 8K_2{}^2K_3{}^3K_1 - 8K_2{}^4K_1{}^2 + 22K_3{}^3K_1{}^3 - 6K_2{}^2K_1{}^4 + 2K_1{}^6)}{K_3{}^9K_1{}^3} , \quad (2.14)$$

and

$$B_{\dot{\zeta}^3}(k_1, k_2, k_3) = \left[6\left(c_s^2 - 1\right) + 8\frac{c_3}{c_s^2}\right]\Delta_\zeta^2 \frac{1}{K_3{}^3K_1{}^3} , \quad (2.15)$$

where we have defined

$$\langle\zeta(\vec{k}_1)\zeta(\vec{k}_2)\zeta(\vec{k}_3)\rangle = B(k_1, k_2, k_3)(2\pi)^3\delta(\vec{k}_1 + \vec{k}_2 + \vec{k}_3) , \quad (2.16a)$$

$$B = B_{\dot{\zeta}(\partial_i\zeta)^2} + B_{\dot{\zeta}^3} , \quad (2.16b)$$

$$K_1 \equiv k_1 + k_2 + k_3 , \qquad K_2 \equiv (k_1k_2 + k_2k_3 + k_3k_1)^{1/2} , \qquad K_3 \equiv (k_1k_2k_3)^{1/3} . \quad (2.16c)$$

Defining $x_i = k_i/k_1$ and changing variables, we find

$$\langle\zeta^3(\vec{x} = 0)\rangle = \int \frac{\mathrm{d}^3k_1}{(2\pi)^3} \frac{1}{k_1^3} \frac{3!}{(2\pi)^2} \int_{1/2}^1 \mathrm{d}x_2\, x_2 \int_{1-x_2}^{x_2} \mathrm{d}x_3\, x_3\, B(1, x_2, x_3)$$

$$= \frac{\log aH/K}{2\pi^2} \times \frac{\Delta_\zeta^2}{16\pi^2}\left(\left(1 - \frac{1}{c_s^2}\right)(9 + 3c_s^2) + \frac{c_3}{c_s^2}\right) , \quad (2.17)$$

where we have simply introduced a hard UV cutoff $k_1 = aH$ and an IR cutoff as $k_1 = K$ to regulate the log-divergence. This simple regulator breaks the symmetries of dS, and so we provide App. A.2, which shows how to derive the same result using the symmetry preserving dynamical dimensional regularization approach.

The coefficient $\gamma_3$ is determined from the factor multiplying the log:

$$\gamma_3 = \frac{\Delta_\zeta^2}{32\pi^4}\left(\left(1 - \frac{1}{c_s^2}\right)(9 + 3c_s^2) + \frac{c_3}{c_s^2}\right) , \quad (2.18)$$

so that

$$\frac{\partial}{\partial\mathtt{t}}P_{\mathrm{NG}}(\zeta, \mathtt{t}) = \left(\frac{\Delta_\zeta}{8\pi^2}\frac{\partial^2}{\partial\zeta^2} - \frac{\Delta_\zeta^2}{192\pi^4}\left(\left(1 - \frac{1}{c_s^2}\right)(9 + 3c_s^2) + \frac{c_3}{c_s^2}\right)\frac{\partial^3}{\partial\zeta^3}\right)P_{\mathrm{NG}}(\zeta, \mathtt{t}) . \quad (2.19)$$

This result is consistent with the interpretation that it is a small non-Gaussian correction: a typical fluctuation in the Gaussian limit is $\zeta \simeq \Delta_\zeta^{1/2}$, so if we assume $\partial/\partial\zeta \sim \Delta_\zeta^{-1/2}$, the first term is $\mathcal{O}(1)$ and the second term is $\mathcal{O}(\Delta_\zeta^{1/2}/c_s^2)$. We can rewrite this estimate in

terms of the cutoff scale, using the relation $\Lambda = f_\pi c_s$:

$$\Delta_\zeta = \frac{1}{2}\frac{H^4}{f_\pi^4} \qquad \Rightarrow \qquad \gamma_3 \frac{\partial^3}{\partial \zeta^3} \simeq \frac{\Delta_\zeta^{1/2}}{c_s^2} = \frac{1}{\sqrt{2}}\frac{H^2}{\Lambda^2} \; . \tag{2.20}$$

This tells us that for typical fluctuations, the higher order corrections are suppressed by $H^2/\Lambda^2$ as one would expect.

# 3  Eternal Inflation and Non-Gaussian Tails

Now that we have the leading corrections to the Fokker-Planck equation in the presence of a non-trivial bispectrum, we want to apply this formalism to see how it impacts the onset of eternal inflation and the implications for the de Sitter entropy. Even without appealing to a microscopic description, we can define an order parameter for the end of inflation $\phi$. Within the EFT of Inflation, there is a natural choice [29]

$$\phi \equiv f_\pi^2(t + \pi) \simeq \frac{f_\pi^2}{H}(\mathtt{t} - \zeta) \; , \tag{3.1}$$

where $\pi$ is the Goldstone boson of the EFT of Inflation (see Appendix A.1 for review) defined such that $\zeta = -H\pi + \mathcal{O}(\epsilon\pi^2)$, where $\epsilon$ is the slow roll parameter, and $f_\pi^2$ is the decay constant for $\pi$. By construction $\langle \dot{\phi} \rangle = f_\pi^2$. Since we will be working in the limit $\epsilon \to 0$, we can treat Eq. (3.1) as an exact relation to define $\zeta$ in terms of $\phi$:

$$\zeta \equiv \mathtt{t} - \frac{H}{f_\pi^2}\phi \; . \tag{3.2}$$

Here $\phi$ will be the field that defines the end of inflation so that $\phi \in (-\infty, 0)$ corresponds to the inflationary regime with inflation ending when $\phi = 0$. (Ending inflation at $\phi = 0$ simplifies expressions, but of course nothing can depend on this arbitrary choice).

To set the stage, we will review how one determines the onset of eternal inflation in the Gaussian case, $\gamma_3 = 0$. The evolution equation for $\zeta$ is

$$\frac{\partial}{\partial \mathtt{t}}P_{\mathrm{G}}(\zeta, \mathtt{t}) = \frac{\Delta_\zeta}{4\pi^2}\frac{\partial^2}{\partial \zeta^2}P_{\mathrm{G}}(\zeta, \mathtt{t}) \; , \tag{3.3}$$

whose solutions are given by a Gaussian:

$$P_{\mathrm{G}}(\zeta, \mathtt{t}, \zeta_0) = \frac{1}{\sqrt{2\pi\sigma^2 \mathtt{t}}}e^{-(\zeta-\zeta_0)^2/(2\sigma^2 \mathtt{t})} \; , \tag{3.4}$$

for any choice of the constant $\zeta_0$, and with $\sigma^2 \equiv \Delta_\zeta/(2\pi^2)$. We impose the initial condition $P_{\mathrm{G}}(\zeta, \mathtt{t} = 0, \zeta_i) = \delta(\zeta - \zeta_i)$ so that $\zeta = \zeta_i > 0$ ($\phi < 0$) at $\mathtt{t} = 0$ in order to be consistent

with Eq. (3.2). Since inflation ends when $\phi \geq 0$, we set $P(\phi[\zeta] \geq 0; \mathtt{t}) = 0$ by hand. However, we must also impose the boundary condition that $P_{\mathrm{G}}(\zeta, \mathtt{t}, \zeta_i)$ is continuous at $\phi[\zeta] = 0$. Note that every choice of $\zeta_0$ in the solution Eq. (3.4) gives a $\delta$-function $\delta(\zeta - \zeta_0)$ at $\mathtt{t} = 0$. In order to impose our boundary condition at $\phi = 0$, we must add additional solutions in the region $\phi[\zeta] < 0$ with different values of $\zeta_0 < 0$ so that they naively produce a $\delta$-function for $\phi > 0$ at $\mathtt{t} = 0$. However, since we are imposing $P_{\mathrm{G}}(\phi > 0, \mathtt{t}) = 0$ by hand, adding these additional terms remains consistent with our initial conditions. A natural guess is that the solution takes the form

$$P_{\mathrm{G}}(\phi[\zeta] < 0, \mathtt{t}, \zeta_i) = \frac{1}{\sqrt{2\pi\sigma^2 \mathtt{t}}} \left[ e^{-(\zeta-\zeta_i)^2/(2\sigma^2 \mathtt{t})} - e^{-4\zeta_i/(2\sigma^2)} e^{-(\zeta+\zeta_i)^2/(2\sigma^2 \mathtt{t})} \right] , \qquad (3.5)$$

where $\phi = 0$ corresponds to $\zeta = \mathtt{t}$. This way of imposing the boundary conditions is typically called the method of images.

Now that we have the probability distribution, we can apply it to compute the onset of eternal inflation. Following [49], the probability that reheating occurs at time $\mathtt{t}$ is determined by

$$p_{\mathrm{R,G}}(\mathtt{t}) = -\frac{\mathrm{d}}{\mathrm{d}\mathtt{t}} \int_{-\infty}^{0} \mathrm{d}\phi \, P_{\mathrm{G}}(\phi; \mathtt{t}) \propto e^{-\mathtt{t}/(2\sigma^2)} , \qquad (3.6)$$

where we used the Fokker-Planck equation Eq. (3.3) and integrated by parts. From here we can calculate the average volume of the reheating surface,

$$\langle V \rangle_{\mathrm{G}} = L^3 \int_0^{\infty} \mathrm{d}\mathtt{t}\, e^{3\mathtt{t}} p_{\mathrm{R,G}}(t) \simeq L^3 \int_0^{\infty} \mathrm{d}\mathtt{t}\, e^{\mathtt{t}(3-1/(2\sigma^2))} . \qquad (3.7)$$

where $L^3$ is the size of the initial patch at $\mathtt{t} = 0$. The onset of eternal inflation occurs when this quantity diverges:

$$\sigma^2 = \frac{\Delta_\zeta}{2\pi^2} > \frac{1}{6} . \qquad (3.8)$$

In canonical slow-roll inflation, the perturbative description remains weakly coupled up to the phase transition and therefore this determination of the critical value of $\Delta_\zeta$ is meaningful [49].

Now let us repeat this analysis for theories with primordial non-Gaussianity, $\gamma_3 \neq 0$. The evolution is described by (see Eq. (2.10))

$$\frac{\partial}{\partial \mathtt{t}} P_{\mathrm{NG}}(\zeta, \mathtt{t}) = \frac{\sigma^2}{2} \frac{\partial^2}{\partial \zeta^2} P_{\mathrm{NG}}(\zeta, \mathtt{t}) - \frac{\gamma_3}{3!} \frac{\partial^3}{\partial \zeta^3} P_{\mathrm{NG}}(\zeta, \mathtt{t}) . \qquad (3.9)$$

Building off the solution in Eq. (3.4), we can make the ansatz for the solution to this modified Fokker-Planck equation:

$$P_{\mathrm{NG}}(\zeta, \mathtt{t}, \zeta_0) = \exp\left( \frac{\kappa(\mathtt{t})}{3!} \frac{\partial^3}{\partial \zeta^3} \right) P_{\mathrm{G}}(\zeta, \mathtt{t}, \zeta_0) . \qquad (3.10)$$

Substituting this ansatz into Eq. (3.9) gives

$$\frac{\mathrm{d}}{\mathrm{dt}}\kappa(\mathsf{t}) = -\gamma_3 \rightarrow \kappa(\mathsf{t}) = -\gamma_3\mathsf{t} + \kappa_0 \ . \tag{3.11}$$

We again impose the initial condition at $\mathsf{t} = 0$, $\zeta = \zeta_i < 0$, and $P(\phi \geq 0) = 0$, so the solution takes the form

$$P_{\mathrm{NG}}(\zeta, \mathsf{t}, \zeta_i) = \exp\left(-\frac{\gamma_3\mathsf{t}}{3!}\frac{\partial^3}{\partial\zeta^3}\right) P_{\mathrm{G}}(\zeta, \mathsf{t}, \zeta_i) + \text{images} \ , \tag{3.12}$$

where the images are solutions with $\zeta_0 > 0$. While this can be solved in principle, a closed form solution to these equations is both unnecessary and beyond our scope. Specifically, the phase transition is determined the behavior at $\phi = 0$ or $\zeta \to \mathsf{t}$ in the limit $\mathsf{t} \to \infty$. In this limit, we have

$$\frac{\partial^n}{\partial\zeta^n}P_{\mathrm{G}}(\zeta, \mathsf{t}, \zeta_0)\Big|_{\zeta=\mathsf{t}} = \left(\frac{(-1)^n}{\sigma^{2n}} + \mathcal{O}(\mathsf{t}^{-1})\right) P_{\mathrm{G}}(\zeta, \mathsf{t}, \zeta_0) \tag{3.13}$$

so that the Gaussian behaves as an eigenfunction of the derivative operator in the $\mathsf{t} \to \infty$ limit. In this regime, the probability distribution for $\zeta$ becomes

$$\exp\left(-\frac{\gamma_3\mathsf{t}}{3!}\frac{\partial^3}{\partial\zeta^3}\right) P_{\mathrm{G}}(\zeta; \mathsf{t}, \zeta_0) \rightarrow \exp\left[\frac{\gamma_3}{3!}\mathsf{t}\left(\frac{(\zeta-\zeta_0)}{\sigma^2\mathsf{t}}\right)^3 - \frac{(\zeta-\zeta_0)^2}{2\sigma^2\mathsf{t}}\right] \ . \tag{3.14}$$

This solution is also derived in App. B using the method of steepest descents.

This probability distribution for $\zeta$ tells us that the large $\mathsf{t}$ behavior for the probability of reheating is

$$p_{\mathrm{R,NG}}(\mathsf{t}) \propto \exp\left[-\mathsf{t}\left(\frac{1}{2\sigma^2} - \frac{\gamma_3}{3!}\frac{1}{\sigma^6}\right)\right] \ . \tag{3.15}$$

Repeating the same argument from above to derive the onset of eternal inflation, we see that $\langle V \rangle_{\mathrm{NG}}$ diverges when

$$\frac{1}{2\sigma^2} - \frac{\gamma_3}{3!}\frac{1}{\sigma^6} < 3 \ . \tag{3.16}$$

Note that this result depends on the sign of $\gamma_3$, which is not fixed. Using the explicit form of $\gamma_3$ given in Eq. (2.18) and $\sigma^2 = \Delta_\zeta/(2\pi^2)$, eternal inflation occurs when

$$\frac{1}{2} - \frac{1}{48}\left(\left(1 - \frac{1}{c_s^2}\right)(9 + 3c_s^2) + \frac{c_3}{c_s^2}\right) < \frac{3\Delta_\zeta}{2\pi^2} \ . \tag{3.17}$$

At this point, we notice something surprising. One might have expected that the Gaussian term would dominate when $H^2/\Lambda^2 \ll 1$. However, if we recall that $\Lambda = f_\pi c_s$, we

can rewrite this expression as

$$\frac{1}{2} - \frac{1}{48}\frac{f_\pi^2}{\Lambda^2}\left(\left(c_s^2 - 1\right)\left(9 + 3c_s^2\right) + c_3\right) < \frac{3\Delta_\zeta}{2\pi^2} \ . \tag{3.18}$$

When computing the onset of eternal inflation, we see the corrections scale as $f_\pi^2/\Lambda^2 \gg H^2/\Lambda^2$. Taken at face value, this implies that for $c_s = 1$ ($c_s \ll 1$), eternal inflation occurred in our universe for $c_3 < 24$ ($c_3 < -9$). In Fig. 2, we compare this region to current observational constraints from Planck denoted by the red contours in Fig. 2 on the parameter space (taking $\Delta_\zeta \simeq 0$), in terms of $c_s$ and

$$\bar{\alpha}_1 \equiv -\frac{4}{3}\frac{c_3}{c_s^2} - \frac{1}{2}\frac{(1 - c_s^2)^2}{c_s^2} \ . \tag{3.19}$$

The figure also shows conservative bounds on these parameters from perturbative unitarity at horizon crossing, derived in [35] (see Appendix A.3 for a review of perturbative unitarity constraints on the EFT of Inflation).

Since the correction we calculated scales as $f_\pi^2/\Lambda^2$ it is natural to guess we have become sensitive to the cutoff scale for the EFT of Inflation. This suggests that we would become sensitive to even higher derivative corrections. We can estimate the size of these terms by dimensional analysis, using their relation to the connected correlators of $\zeta$. Using our normalization of higher dimension operators in terms of $\Lambda$, we have

$$\gamma_n \simeq \sigma^n \left(\frac{H}{\Lambda}\right)^{2n-4} \ . \tag{3.20}$$

Extending the ansatz in Eq. (3.10) to include higher derivatives, we find

$$P(\zeta; \mathtt{t}, \zeta_0) = \exp\left(\sum_{n>2}(-1)^n \frac{\gamma_n \mathtt{t}}{n!}\frac{\partial^n}{\partial\zeta^n}\right) P_{\mathrm{G}}(\zeta; \mathtt{t}, \zeta_i) + \text{images} \ , \tag{3.21}$$

so that

$$p_{\mathrm{R}}(\mathtt{t}) \propto \exp\left[-\mathtt{t}\left(\frac{1}{2\sigma^2} + \sum_{n>2}\frac{\gamma_n}{n!}\frac{1}{\sigma^{2n}}\right)\right] \ . \tag{3.22}$$

Again, we see that in the $\mathtt{t} \to \infty$ limit, all the $\gamma_n$ corrections contribute to coefficient of the exponential decay but do not change powers of $\mathtt{t}$ in the exponent. As a result, the reheating volume diverges when

$$\frac{1}{2} + \sum_{n>2}(-1)^n \frac{\gamma_n}{n!}\frac{1}{\sigma^{2n-2}} < 3\sigma^2 \ . \tag{3.23}$$

Now we notice that the $n^{\text{th}}$ term is the sum is

$$\frac{\gamma_n}{n!} \frac{1}{\sigma^{2n-2}} \simeq \frac{1}{n!} \frac{1}{\sigma^{n-2}} \left(\frac{H}{\Lambda}\right)^{2n-4} \frac{1}{\Delta_\zeta^{(n-1)/2}} \simeq \left(\frac{f_\pi}{\Lambda}\right)^{2n-4} . \tag{3.24}$$

Therefore, the series is under control for typical couplings when $\Lambda > f_\pi$.

On the other hand, when $\Lambda < f_\pi$, it is possible, in principle, to tune the coefficients of the higher order terms so that $\gamma_3$ is the dominant contribution. Yet, the fact that our results naturally organize into an expansion in $f_\pi/\Lambda$ suggests that something more drastic is occurring in the parameter space where $\Lambda < f_\pi$ that cannot be resolved by fine tuning. We will revisit this interpretation in Sec. 5.

# 4 The de Sitter Entropy and Microstate Counting

The interpretation of these corrections in the context of eternal inflation becomes even more more drastic when we apply them [2, 67, 68] to our interpretation of the de Sitter entropy [66],

$$S_{\text{dS}} = \frac{\pi}{H^2 G_{\text{N}}} = \frac{8\pi^2 M_{\text{pl}}^2}{H^2} , \tag{4.1}$$

where $G_{\text{N}}$ is Newton's constant. During inflation, the de Sitter entropy is slowly changing as $H(t)$ decreases, such that

$$\frac{\mathrm{d}S_{\text{dS}}}{\mathrm{d}t} = \frac{\mathrm{d}S_{\text{dS}}}{H\mathrm{d}t} = -\frac{16\pi^2 M_{\text{pl}}^2 \dot{H}}{H^4} = \frac{4\pi^2}{c_s \Delta_\zeta} . \tag{4.2}$$

In analogy with the entropy of a black hole, it is natural to interpret this entropy as reflecting a finite number of degrees of freedom describing the microphysics of (quasi) de Sitter space. One crude test of this hypothesis is to compare the de Sitter entropy to the entropy of the fluctuations that are observable after inflation ends, following [2]. The number of Fourier modes that are being "created" (*i.e.*, crossing the horizon) per e-fold is simply the expansion rate

$$\frac{\mathrm{d}\log N_{\text{modes}}}{\mathrm{d}t} = 3 . \tag{4.3}$$

If the de Sitter entropy is to be interpreted as resulting from the size of the Hilbert space describing the modes that live in de Sitter, $S_{\text{dS}} \propto \log N_{\text{states}}$, then we should be prevented from observing more than a de Sitter entropy's worth of Fourier modes, so that $N_{\text{modes}} < N_{\text{states}}$, which implies

$$\int \mathrm{d}t \frac{\log N_{\text{modes}}}{\mathrm{d}t} < \int \mathrm{d}t \frac{\mathrm{d}S_{\text{dS}}}{\mathrm{d}t} . \tag{4.4}$$

Our general expectation is that $N_\text{modes} \ll N_\text{states}$ since the semi-classical fluctuations should capture only a small fraction of the gravitational microstates.

In the models of interest here, the integrands are nearly constant so that Eq. (4.4) holds at the level of the integrand:

$$\frac{\mathrm{d}\log N_\text{modes}}{\mathrm{dt}} = 3 < \frac{\mathrm{d}S_\text{dS}}{\mathrm{dt}} = \frac{4\pi^2}{c_s \Delta_\zeta} \ . \tag{4.5}$$

Naively, one can imagine violating this interpretation by taking

$$\Delta_\zeta \overset{?}{>} \frac{4\pi^2}{3c_s} \ . \tag{4.6}$$

However, to derive a contradiction, it should be unambiguous that all $N_\text{modes}$ are independent and observable. This would be verifiable if inflation ended everywhere in the universe, allowing us a vantage point from which to reconstruct all of inflation. However, if inflation never ends, $i.e.$, we are eternally inflating, then these modes are not accessible to an observer. In canonical slow-roll inflation ($c_s = 1$ and $c_3 = 0$), the onset of eternal inflation was determined in Eq. (3.8). Therefore, a finite period of inflation always satisfies the inequality

$$\Delta_\zeta < \frac{\pi^2}{3} < \frac{4\pi^2}{3} \ . \tag{4.7}$$

As a result, we never encounter a regime where more than $e^{S_\text{dS}}$ modes are produced while maintaining control of the background in canonical slow-roll inflation.

In the presence of non-Gaussianity, the onset of eternal inflation is modified, potentially allowing a contradiction with this interpretation of the de Sitter entropy. In particular, demanding a finite inflationary volume while violating our de Sitter entropy bound is possible when

$$\frac{\pi^2}{3} - \frac{\pi^2}{72}\left(\left(1 - \frac{1}{c_s^2}\right)(9 + 3c_s^2) + \frac{c_3}{c_s^2}\right) > \Delta_\zeta > \frac{4\pi^2}{3c_s} \ . \tag{4.8}$$

When $c_s \ll 1$, the left hand side of this equality scales as $1/c_s^2$, which easily allows a window where $\Delta_\zeta > 4\pi^2/(3c_s)$ without transitioning to eternal inflation. If we set $c_3 = 0$, this equality can be satisfied for any

$$c_s < 0.095 \ . \tag{4.9}$$

When $c_3 \neq 0$ and $c_s \ll 1$, we can satisfy the inequality for $c_3 < 9$.

Rather than seeing this as a breakdown of the relation between the de Sitter entropy and the microstate counting, it is natural to interpret this as a breakdown of the EFT of Inflation. The likely possibility is that $c_s < 0.095$ is in the strongly coupled regime of the EFT of Inflation, telling us that we cannot trust the calculation of the onset of eternal

inflation. Concretely, we can again rewrite this equality in terms of $\Lambda = c_s f_\pi$ as

$$\frac{\pi^2}{3} + \frac{\pi^2}{72}\frac{f_\pi^2}{\Lambda^2}\left(\left(1 - c_s^2\right)\left(9 + 3c_s^2\right) - c_3\right) > \Delta_\zeta > \frac{4\pi^2}{3c_s} \ . \tag{4.10}$$

We can again only satisfy this inequality when $\Lambda^2 < f_\pi^2$, and in the case $c_s \ll 1$ we require $\Lambda^2 \ll f_\pi^2$. However, eternal inflation occurs when $\Delta_\zeta > 4\pi^2/(3c_s)$:

$$\frac{H^4}{f_\pi^4} > \frac{8\pi^2}{3c_s} \ . \tag{4.11}$$

This is only satisfied for $f_\pi < H$ and therefore the regime of interest is where $\Lambda^2 \ll H^2$. This strongly suggests that we cannot see more than a de Sitter entropy's worth of modes in the regime that is under control within the EFT of Inflation. We might even interpret $c_s > 0.095$ (when $\Delta_\zeta > \frac{4\pi^2}{3c_s}$) as a bound on the regime of validity of the EFT defined by the de Sitter entropy. We compare this to the bound from naively applying perturbative unitarity in Appendix A.3 and find good agreement. One may hope that a further exploration of the de Sitter entropy will bound the range of parameters in the EFT of Inflation directly from the cosmological background, complementing other approaches more similar to QFT in flat space [32, 101, 102].

## 5  On the Breakdown of Stochastic Inflation

By direct calculation, we have shown that the presence of primordial non-Gaussianity leads to a series of large corrections that can dramatically modify the onset of eternal inflation when $\Lambda < f_\pi$. Our goal here is to make the case that this should be interpreted as a breakdown of Stochastic Inflation akin to the breakdown of the EFT of Inflation in the strong coupling regime. This should not prevent us from calculating the phase transition in a UV complete model. We can understand both issues by returning to the origins of Stochastic Inflation.

The Stochastic framework follows as a consequence of general Markovian evolution. For a scalar field $\phi$, this evolution is described by

$$\frac{\partial}{\partial t}P(\phi, t) = \int \mathrm{d}\Delta\phi\Big[P(\phi - \Delta\phi, t)W(\phi|\phi - \Delta\phi) - P(\phi, t)W(\phi + \Delta\phi|\Delta\phi)\Big] , \tag{5.1}$$

where $W(\phi|\phi')$ is the transition amplitude for the field to jump from $\phi'$ to $\phi$ during the time $\mathrm{d}t$. If these transition amplitudes are sufficiently "local," we can Taylor expand Eq. (5.1) to get

$$\frac{\partial}{\partial t}P(\phi, t) = \sum_{n=1}^{\infty}\frac{1}{n!}\frac{\partial^n}{\partial\phi^n}\,\Omega_n(\phi)P(\phi, t) , \tag{5.2}$$

where

$$\Omega_n(\phi) \equiv \int d\Delta\phi \left(-\Delta\phi\right)^n \widetilde{W}(\Delta\phi, \phi), \tag{5.3}$$

and $\widetilde{W}(y,x) \equiv W(x+y|x)$. For approximately Gaussian transition amplitudes, the moments of the distribution should be well defined, leading to a reasonable derivative expansion. Indeed, for the case of $\lambda\phi^4$ theory [51] and inflation (Sec. 2 above), we have verified that these coefficients are calculable by explicitly evaluating them.

Scalar metric fluctuations $\zeta$ are constrained by an additional non-linearly realized symmetry (Eq. (2.3)), which enforces that $W(\zeta|\zeta') = W(\zeta - \zeta')$ or $\widetilde{W}(y,x) \equiv \widetilde{W}(y)$. Using the expected scaling behavior for $\gamma_n$ given in Eq. (3.20), we write

$$\gamma_n = g_n\sigma^n \left(\frac{H}{\Lambda}\right)^{2n-4}, \tag{5.4}$$

so that $g_n = \mathcal{O}(1)$ and

$$\frac{\partial}{\partial t}P(\zeta,t) = \left(\frac{\sigma^2}{2}\frac{\partial^2}{\partial\zeta^2} + \frac{\Lambda^4}{H^4}\sum_{n=3}^{\infty}(-1)^n\frac{1}{n!}g_n\left(\sigma\frac{H^2}{\Lambda^2}\frac{\partial}{\partial\zeta}\right)^n\right)P(\zeta,t), \tag{5.5}$$

This is the scaling behavior we would get from a transition amplitude of the form

$$W(\Delta\zeta) = \frac{1}{\sqrt{2\pi^2\sigma^2}}\exp\left[-\frac{(\Delta\zeta)^2}{2\sigma^2}\left(1 + \sum_{n>2}(-1)^n g_n\left(\frac{2\pi f_\pi^2(\Delta\zeta)}{\Lambda^2}\right)^{n-2}\right)\right], \tag{5.6}$$

where $\Delta\zeta \equiv \zeta - \zeta'$. This series expansion will break down when

$$\Delta\zeta > \frac{\Lambda^2}{2\pi f_\pi^2}, \tag{5.7}$$

or, using $\Delta\zeta = H\Delta\phi/f_\pi^2$,

$$H\Delta\phi > \frac{\Lambda^2}{2\pi}. \tag{5.8}$$

Notice that the expression on the left is $H\Delta\phi \simeq \dot{\phi}$. It is natural to interpret the large changes in the field over a Hubble time as a probe of the high energy limit of the theory $E \gg H$, which explains why we encounter the EFT cutoff scale $\Lambda$.

If the breakdown is indeed due to strong coupling within the EFT, we would expect that a large correction to the onset of eternal inflation would coincide with violations of perturbative unitarity at energies $E = f_\pi$. Since $f_\pi \gg H$, we can approximate the subhorizon region as flat space and can calculate the the partial wave amplitudes for two-to-two scattering of $\zeta$. These amplitudes are provided in App. A.3 along with their associated perturbative unitarity bounds. If we take $c_s = 1$, then $s$-wave scattering in the

center of mass frame with incoming energies of $E = f_\pi$ is consistent with perturbativity when

$$-1.85 \leq c_3 \leq 0.85 \ . \tag{5.9}$$

For comparison, we saw that the non-Gaussian corrections naively imply an infinite reheating volume when $c_3 < 24$ (again for $c_s = 1$). In this sense, the breakdown of our intuition regarding eternal inflation is indeed tied to the breakdown of perturbative unitarity at $E = f_\pi$.

Ultimately, the question of whether or not a given model of inflation is eternally inflating should be calculable. However, clearly the method of calculating the correlation functions to determine equations of Stochastic Inflation is insufficient. Furthermore, nothing about the calculation of the individual correlation functions will change if we work in the UV completion (rather than the EFT). This is particularly clear when Stochastic Inflation is expressed in terms of $\zeta$, so that all the coefficients are constants and can be calculated from the perturbative correlation functions.

## 5.1   Breakdown in DBI

For concreteness, DBI inflation [103] provides a useful analogy from which we can try to understand the breakdown of our calculation [23]. In DBI, the action is given by

$$\mathcal{L} = \Lambda^4 \sqrt{1 + \frac{\partial_\mu \phi \partial^\mu \phi}{\Lambda^4}} - V(\phi) \ . \tag{5.10}$$

The potential $V(\phi)$ generates a rolling field, $\phi = \dot{\phi}(t + \pi)$. Expanding the square root, we find

$$\Lambda^4 \sqrt{1 - \frac{\dot{\phi}^2 (1 + 2\dot{\pi} - \partial_\mu \pi \partial^\mu \pi)}{\Lambda^4}} \ \rightarrow \ \sum_n \frac{1}{n!} M_n^4 (2\dot{\pi} - \partial_\mu \pi \partial^\mu \pi)^n \ , \tag{5.11}$$

where

$$M_n^4 = \Lambda^4 (-1)^n \left( \frac{\dot{\phi}^2}{\Lambda^4} \right)^n \frac{\partial^n}{\partial X^n} \sqrt{1 - X} \Big|_{X = \dot{\phi}^2 / \Lambda^4} \ , \tag{5.12}$$

and

$$\frac{c_s^2}{1 - c_s^2} = 1 - \frac{\dot{\phi}^2}{\Lambda^4} \ , \tag{5.13}$$

so that $c_s \ll 1$ requires $\dot{\phi}^2 \simeq \Lambda^4$. Clearly in that limit, any process that involves a transition $\dot{\phi} \simeq H \Delta \phi > \Lambda^2$ will require the full DBI action. In fact, we see the Taylor expansion of Eq. (5.11) will break down when sooner, when

$$\frac{\dot{\phi}^2}{\Lambda^4} \left( 2 \frac{\dot{\pi}_c}{f_\pi^2} - \frac{\partial_\mu \pi_c \partial^\mu \pi_c}{f_\pi^4} \right) \ll 1 - \frac{\dot{\phi}^2}{\Lambda^4} = c_s^2 \quad \xrightarrow{c_s \ll 1} \quad \frac{\dot{\pi}_c}{f_\pi^2} \ll \frac{c_s^2}{2}, \tag{5.14}$$

where $\pi_c = f_\pi^2 \pi$ is the canonically normalized field in the EFT of Inflation (see Appendix A for review). Since $\pi_c$ scales like the energy of the mode $E$, $\dot\pi_c \simeq E^2$ and, therefore, Eq. (5.14) tells us that our Taylor expansion is only valid for energies $E^2 \ll c_s^2 f_\pi^2/2 = \Lambda^2/2$. The cutoff scale we identified in the EFT is the scale where we can no long Taylor expand the DBI action.

The DBI example naturally suggests the how to resolve this breakdown: we need to calculate the full transition amplitudes using the UV completion.[5] In the case of DBI, we expect that modeling large field transitions requires knowing the complete non-perturbative form of the DBI action. In contrast, for smaller transitions, we can Taylor expand the DBI action to reproduce the same results as we would find using the EFT of Inflation. In practice, DBI may not be the simplest model in which to directly calculate the full transition amplitude, as more weakly coupled UV completions of small $c_s$ may offer some advantages. We leave exploring this interesting direction to future work.

# 6    Conclusions

In this paper, we extended the framework of Stochastic Inflation in single field inflation to include the impact of (equilateral) primordial non-Gaussianity. We showed that the single field consistency conditions demand that these corrections can only include higher derivative terms with constant coefficients. We then calculated a two-loop anomalous dimension in SdSET and used it to determine the cubic derivative correction to Stochastic Inflation.

Using these evolution equations, we set out to calculate the onset of eternal inflation in the presence of non-Gaussian fluctuations. We found that this transition cannot be calculated within the framework of Stochastic Inflation when the EFT of Inflation is not weakly coupled at the scale of the time evolution of the background, $f_\pi = 59H$, which is well above the scale of horizon crossing $H$. For a wide variety of models producing observable non-Gaussianity, the onset of eternal inflation is incalculable and requires appealing to a UV completion of the Stochastic framework and the EFT of Inflation.

We interpret the breakdown of Stochastic Inflation as a sign that the tail of the probability distribution for the scalar fluctuations is a probe of sub-horizon physics during inflation. This conclusion is relevant to other probes of non-Gaussianity from rare fluctuations [106–110], most notably as applied to the formation of primordial black holes (PBHs) [111–114]. Like the onset of eternal inflation, the rate of PBH formation in canonical slow-roll inflation is calculable using Stochastic Inflation, both in the single and multi-field regime (see *e.g.* [115, 116] for discussions of the connection between PBHs and Stochastic Inflation). Nevertheless, generic non-Gaussianity was known to impact these

---

[5]The contributions to Stochastic Inflation in DBI were previous discussed in [104, 105]. The contributions for higher derivatives did not appear in those works and thus they did not find the need for a non-perturbative calculation.

rates of rare fluctuations in ways that might not be calculable in perturbation theory [117]. Our results suggest it is the EFT of Inflation that is breaking down, which implies that one cannot resolve this effect within the EFT itself, *e.g.* by resumming EFT Feynman diagrams.

This work makes a sharp connection between a number of important topics in theoretical and observational cosmology: the regime of validity of cosmological EFTs, primordial non-Gaussianity, probes of the tail of the distribution of scalar fluctuations, eternal inflation, and the de Sitter entropy. A natural next step is to explore the connection between these results in models that are UV completed beyond the cutoff of the EFT of Inflation. Given a concrete model, *e.g.* DBI inflation [103], one could compute the tail of the distribution or the onset of eternal inflation. More generally, de Sitter holography (quantum gravity) also connects many of these topics and offers a parallel and unique perspective on the de Sitter entropy [3, 69–71], non-Gaussianity [90, 118] and (potentially) eternal inflation [119, 120]. Naturally, one would like to understand the breakdown of Stochastic Inflation from a holographic perspective. Our results imply the need for a deeper non-perturbative definition of eternal inflation, which may provide a concrete opportunity to link these (often) distinct approaches to cosmology.

**Acknowledgements**    We are grateful to Daniel Baumann, Tom Hartman, Yiwen Huang, Mehrdad Mirbabayi, Gui Pimentel, Rafael Porto, Chia-Hsien Shen, and Eva Silverstein for helpful discussions. D.G. also thanks the participants of *State of the Quantum Universe* for valuable insights. T.C. is supported by the US Department of Energy, under grant no. DE-SC0011640. D.G. and A.P. are supported by the US Department of Energy under grants DE-SC0019035 and DE-SC0009919.

# Appendices

# A    Calculations for Single Field Inflation

In this appendix, we review the basics of the EFT of Inflation, and review results for the power spectrum and bispectrum that are used for the calculations in the main text. Then in App. A.2, we provide some technical details for how to regulate the key integral that appears in the calculation above using a dimensional regularization like approach that explicitly preserves the symmetries of the problem. We then review the perturbative unitarity constraint derived using flat space amplitudes in App. A.3.

## A.1    EFT of Inflation: Power Spectrum and Bispectrum

The EFT of inflation, in the decoupling limit, is described in terms of a Goldstone boson $\pi$ that non-linearly realizes time translations that are broken by the evolving background.

Since $\pi$ shifts by a constant under time translations, the variable $U = t + \pi(t, \vec{x})$ transforms linearly under time translations. We can then express the action in terms of $U$:

$$S = \int \mathrm{d}t\, \mathrm{d}^3 x \sqrt{-g} \sum_{n=0}^{\infty} \frac{1}{n!} M_n^4(U) \left( \partial_\mu U \partial^\mu U + 1 \right)^n , \qquad (A.1)$$

such that the $n^{\text{th}}$ term is $\mathcal{O}(M_n^4 \pi^n)$ (note we are using the $(- + ++)$ signature for the metric). The coefficients $M_0^4$ and $M_1^4$ are fixed by Einstein's equations (or equivalently by eliminating the tadpole). The resulting quadratic and cubic contributions to the Lagrangian are given by

$$\mathcal{L}_2 = M_{\mathrm{pl}}^2 |\dot{H}| \left( \dot{\pi}^2 - \left( \vec{\nabla}\pi \right)^2 \right) + 2M_2^4 \dot{\pi}^2 = \frac{M_{\mathrm{pl}}^2 |\dot{H}|}{c_s^2} \left( \dot{\pi}^2 - c_s^2 \left( \vec{\nabla}\pi \right)^2 \right) , \qquad (A.2)$$

and

$$\mathcal{L}_3 = \left( 2M_2^4 - \frac{4}{3} M_3^4 \right) \dot{\pi}^3 - 2M_2^4 \dot{\pi} \left( \vec{\nabla}\pi \right)^2 , \qquad (A.3)$$

where $M_2^4 = M_{\mathrm{pl}}^2 |\dot{H}| (1 - c_s^2)/(2c_s^2)$.

Using $\zeta = -H\pi + \mathcal{O}(\epsilon\pi^2)$, where $\epsilon = -\dot{H}/H^2$ is the slow roll parameter, and defining the pion decay constant as $f_\pi^4 \equiv 2M_{\mathrm{pl}}^2 |\dot{H}| c_s$, one finds the power spectrum at zeroth order in slow-roll is

$$\langle \zeta(\vec{k}) \zeta(\vec{k}') \rangle' = \frac{H^4}{2f_\pi^4} \frac{1}{k^3} (2\pi)^3 \delta\left( \vec{k} + \vec{k}' \right) \equiv \frac{\Delta_\zeta}{k^3} (2\pi)^3 \delta\left( \vec{k} + \vec{k}' \right), \qquad (A.4)$$

and the bispectra are [100]

$$B_{\dot{\zeta}(\partial_i \zeta)^2} (k_1, k_2, k_3) = -\frac{1}{4} \left( 1 - \frac{1}{c_s^2} \right) \cdot \Delta_\zeta^2$$
$$\times \frac{\left( 24 K_3^6 - 8 K_2^2 K_3^3 K_1 - 8 K_2^4 K_1^2 + 22 K_3^3 K_1^3 - 6 K_2^2 K_1^4 + 2 K_1^6 \right)}{K_3^9 K_1^3} , \qquad (A.5)$$

and

$$B_{\dot{\zeta}^3} (k_1, k_2, k_3) = 4 \left( 1 - \frac{1}{c_s^2} \right) \left( \tilde{c}_3 + \frac{3}{2} c_s^2 \right) \cdot \Delta_\zeta^2 \cdot \frac{1}{K_3^3 K_1^3} . \qquad (A.6)$$

The coefficient $\tilde{c}_3$ is defined in [121] such that it is related to our $c_3$ by

$$\tilde{c}_3 M_2^4 \equiv M_3^4 \equiv c_3 \frac{f_\pi^4}{c_s^5} \quad \rightarrow \quad \tilde{c}_3 = \frac{2c_3}{(1 - c_s^2) c_s^2} , \qquad (A.7)$$

so that

$$B_{\dot{\zeta}^3} (k_1, k_2, k_3) = \left[ 6 \left( c_s^2 - 1 \right) + 8 \frac{c_3}{c_s^2} \right] \cdot \Delta_\zeta^2 \cdot \frac{1}{K_3^3 K_1^3} . \qquad (A.8)$$

We have also defined $K_i$ the set of symmetry functions of the magnitudes of the momenta $k_{1-3}$,

$$K_1 \equiv k_1 + k_2 + k_3 \ , \qquad K_2 \equiv (k_1 k_2 + k_2 k_3 + k_3 k_1)^{1/2} \ . \qquad K_3 \equiv (k_1 k_2 k_3)^{1/3} \ . \qquad \text{(A.9)}$$

Since the bispectra are necessarily symmetric under permutations of $\vec{k}_i$, it is natural to write the correlators in terms of these symmetric functions. The appearance of poles in $K_3$ is particularly noteworthy, as these are the cosmological avatars of energy conservation (also known as the total energy pole).

## A.2   Dynamical Dimensional Regularization

When working with scalar fields in fixed dS, we were able to regulate divergent integrals in a symmetry preserving way by introducing a mass for the scalars, a procedure we called dynamical dimensional regularization (dyn dim reg) [50]. Our interest here is in regulating divergent integrals involving the adiabatic mode $\zeta$. However, $\zeta$ transforms non-trivially under the non-linearly realized symmetries defined in Eq. (2.3) above. (This explains why correlation functions of the adiabatic mode $\zeta$ are time-independent outside the horizon.) As a result, we cannot regulate divergences by introducing a mass for $\zeta$, without breaking these symmetries. Fortunately, by definition, the background itself breaks time-translations, which will provide us with a way to regulate divergent integrals while respecting the symmetries. Having such a symmetry preserving regulator is important for justifying the consistency of the EFT approach.

To see how dyn dim reg works in this setting, we will recompute the leading quantum-noise term in Stochastic Inflation from correlators of $\zeta$ when $c_s = 1$. For comparison, we performed this calculation using a hard cutoff above, see Eq. (2.11). We start with the quadratic action

$$S = \int \mathrm{d}t\, \mathrm{d}^3 x\, a^3(t) \frac{M_{\mathrm{pl}}^2 \dot{H}(t)}{H^2(t)} \partial_\mu \zeta \partial^\mu \zeta \ . \qquad \text{(A.10)}$$

Here we are making the time dependence of $H(t)$ manifest, since we will use this property to regulate divergences. For a general inflation model, $H(t)$ is some arbitrary function of $t$. If we define some reference time $t_\star$, we can therefore expand $H(t)$ as a power series near $t = t_\star$

$$S = \int \mathrm{d}t\, \mathrm{d}^3 x \frac{M_{\mathrm{pl}}^2 \dot{H}(t_\star)}{H^2(t_\star)} \left(\frac{\tau}{\tau_\star}\right)^{2+2\epsilon+\eta} \left(\zeta'^2 - \partial_i \zeta \partial^i \zeta\right) \ , \qquad \text{(A.11)}$$

where

$$\epsilon \equiv -\frac{\dot{H}}{H^2} \ , \qquad \text{and} \qquad \eta = \frac{\dot{\epsilon}}{H\epsilon} \ , \qquad \text{(A.12)}$$

$\tau \simeq -1/(aH)$ is the conformal time, and we used

$$\tau \frac{\mathrm{d}}{\mathrm{d}\tau} \log\left(\frac{H^4(t)}{M_{\mathrm{pl}}^2|\dot{H}(t)|}\right) = \tau \frac{\mathrm{d}}{\mathrm{d}\tau} \log\left(\frac{H^2(t)}{M_{\mathrm{pl}}^2\epsilon(t)}\right) = -2\epsilon - \eta \ . \tag{A.13}$$

The resulting power spectrum is

$$\langle \zeta(\vec{k})\zeta(\vec{k}')\rangle' = \frac{H^4(t_\star)}{4M_{\mathrm{pl}}^2|\dot{H}(t_\star)|} \frac{1}{k^3} \left(\frac{k_\star}{k}\right)^{2\epsilon+\eta} \tag{A.14}$$

where $k_\star\tau_\star = -1$. We can use this to repeat the one-loop calculation we performed in Eq. (2.11) using a hard cutoff:

$$\zeta^n \to \zeta^{n-2} \frac{n(n-1)}{2} \int \frac{\mathrm{d}^3k}{(2\pi)^3} \frac{H^4(t_\star)}{4M_{\mathrm{pl}}^2|\dot{H}(t_\star)|} \frac{1}{k^3} \left(\frac{k}{k_\star}\right)^{-2\epsilon-\eta}$$

$$= \zeta^{n-2} \frac{n(n-1)}{4\pi^2} \frac{H(t_\star)^4}{M_{\mathrm{pl}}^2|\dot{H}(t_\star)|} \left(-\frac{1}{2\epsilon+\eta} + \log K\tau_\star\right) \ , \tag{A.15}$$

where $K$ is an IR regulator[6]. To regulate the divergence, we introduce the normalized operator and the counterterm $Z - 1$ in the minimal subtraction scheme:

$$\zeta_R^{n-2} = Z\zeta^{n-2} \qquad \text{with} \qquad Z = 1 + \frac{1}{2\epsilon+\eta} \frac{n(n-1)}{4\pi^2} \frac{H(t_\star)^4}{M_{\mathrm{pl}}^2|\dot{H}(t_\star)|} \ . \tag{A.16}$$

From here we can determine the dynamical RG by enforcing that our predictions are independent of $aH(t_\star)$, which yields

$$\frac{\mathrm{d}\log Z}{\mathrm{d}\log aH(t_\star)} = -\frac{n(n-1)}{4\pi^2} \frac{H(t_\star)^4}{M_{\mathrm{pl}}^2|\dot{H}(t_\star)|} \ . \tag{A.17}$$

For comparison, we can repeat the calculation using a hard UV cutoff $aH$ with $\epsilon, \eta \to 0$:

$$\zeta^n \to \zeta^{n-2} \times \frac{n(n-1)}{4\pi^2} \frac{H^4}{4M_{\mathrm{pl}}^2|\dot{H}|} \log aH/K \ . \tag{A.18}$$

This yields a counterterm

$$Z = 1 - \frac{n(n-1)}{4\pi^2} \frac{H^4}{4M_{\mathrm{pl}}^2|\dot{H}} \log aH(t_\star)/K \ , \tag{A.19}$$

---

[6]This result is similar to a $1/(n_s - 1)$ pole found in the one-loop power spectrum in [122]. The appearance of these inverse powers of slow-roll parameters in loop calculations is, a priori, not necessarily problematic as they signal the need for dynamical RG in the sense we describe here.

from which we can recover

$$\frac{\mathrm{d}\log Z}{\mathrm{d}\log aH(t_\star)} = -\frac{n(n-1)}{4\pi^2}\frac{H^4}{M_{\mathrm{pl}}^2|\dot{H}|} \ .$$

(A.20)

In this sense, using the hard cutoff at one-loop will reproduce the results of a more careful treatment with dynamical dim reg and minimal subtraction. Precisely the same approach can be applied to regulate the two loop integral in the main text as well.

## A.3 Perturbative Unitarity from Scattering

To understand the results of this paper, it is helpful to understand the energy scales where various processes become important. With the introduction of a non-trivial speed of sound, understanding the physical scales in the problem becomes more challenging, as the distinction between a momentum and energy scale is important. To simplify the problem, we can rescale the spatial coordinates to put time and space on the same footing

$$\tilde{x}^i = x^i/c_s \ , \qquad \tilde{L} = c_s^3 \mathcal{L} \ , \qquad \pi = f_\pi^2 \pi_c \ , \qquad M_n^4 \equiv c_n \frac{f_\pi^4}{c_s^{2n-1}} \ , \qquad \Lambda = f_\pi c_s \ , \quad \text{(A.21)}$$

so that $c_2 \equiv \frac{1}{4}(1 - c_s^2)$. After rescaling, it is convenient to organize the action in terms of the artificially Lorentz invariant derivatives $\tilde{\partial}_\mu$ and time derivatives, so that the action takes the form

$$\tilde{\mathcal{L}} = -\frac{1}{2}\left(\tilde{\partial}\pi_c\right)^2 + \frac{1}{\Lambda^2}\left[\alpha_1\dot{\pi}_c^3 - \alpha_2\dot{\pi}_c\left(\tilde{\partial}\pi_c\right)^2\right] + \frac{1}{\Lambda^4}\left[\beta_1\dot{\pi}_c^4 - \beta_2\dot{\pi}_c^2\left(\tilde{\partial}\pi_c\right)^2 + \beta_3\left(\tilde{\partial}\pi_c\right)^4\right] \ ,$$

(A.22)

where

$$\alpha_1 \equiv -2c_2\left(1 - c_s^2\right) - \tfrac{4}{3}c_3 \ , \quad \alpha_2 \equiv 2c_2 \ ,$$

$$\beta_1 \equiv \tfrac{1}{2}c_2\left(1 - c_s^2\right)^2 + 2c_3\left(1 - c_s^2\right) + \tfrac{2}{3}c_4 \ , \quad \beta_2 \equiv -c_2\left(1 - c_s^2\right) - 2c_3 \ , \quad \beta_3 \equiv \tfrac{1}{2}c_2 \ .$$

(A.23)

Crucially, we notice that the quadratic action is Lorentz invariant in terms of the $\tilde{x}$ variables. The scattering amplitude for $2 \to 2$ scattering in the center of mass frame is [32]

$$M(s, \theta) = \left(-\frac{9}{4}\alpha_1^2 - 4\alpha_2^2 - 6\alpha_1\alpha_2 + \frac{3}{2}\beta_1 + 2\beta_2 + (3 + \cos^2\theta)\beta_3\right)\frac{s^2}{\Lambda^4} \ .$$

(A.24)

Now if we define the partial wave expansion of the amplitude as

$$M(s, \theta) \equiv 16\pi\sum_\ell(2\ell + 1)a_\ell(s)P_\ell(\cos\theta) \ ,$$

(A.25)

then $|a_\ell| \leq 1/2$ in order for the partial waves to be consistent with the optical theorem in perturbation theory, *i.e.*, the theory satisfies the constraint of perturbative unitarity. Integrating the amplitude over $\cos\theta$ we find

$$a_0(s) = \frac{1}{192\pi} \left(-3(3\alpha_1 + 4\alpha_2)^2 + 18\beta_1 + 24\beta_2 + 40\beta_3\right) \frac{s^2}{\Lambda^4} \,, \tag{A.26a}$$

$$a_2(s) = \frac{\beta_3}{120\pi} \frac{s^2}{\Lambda^4} \,. \tag{A.26b}$$

For $a_0$, perturbative unitarity places a bound on a complicated linear combination of $c_s$, $c_3$ and $c_4$. This degeneracy can be broken by considering scattering in boosted frame. In contrast, if we demand $|a_2| < 1/2$ when the external energies are both $f_\pi$ so that $s = 4f_\pi^2$, then we find that $c_s > 0.31$ if perturbative unitarity holds [32].

These bounds can be strengthen by considering scattering beyond the center of mass frame. This is analysis was performed in [35], leading to the exclusion in terms of $c_s$ and $\bar\alpha_1$:

$$\bar\alpha_1 = -\frac{4}{3}\frac{c_3}{c_s^2} - \frac{1}{2}\frac{(1-c_s^2)^2}{c_s^2} = \frac{\alpha_1}{c_s^2} \,. \tag{A.27}$$

This is the bound shown in Fig. 2.

**Consistency with the de Sitter Entropy**

With some extrapolation, we can also apply these results to the consistency of the de Sitter entropy discussed in Sec. 4. Concretely, we could observe more than a de Sitter entropy's worth of modes if we could satisfy the inequalities

$$\frac{\pi^2}{3} - \frac{\pi^2}{72} \left(\left(1 - \frac{1}{c_s^2}\right)(9 + 3c_s^2) + \frac{c_3}{c_s^2}\right) > \Delta_\zeta > \frac{4\pi^2}{3c_s} \,. \tag{A.28}$$

This appears to be possible when $c_s < 0.095$, but we interpreted as a breakdown of EFT of inflation as the cutoff dropped below the Hubble scale, $\Lambda < H$. We check this interpretation by comparing it to the perturbative unitarity on $c_s$ of the $d$-wave amplitude, $|a_2(E)| < 1/2$, which implies [32]

$$\frac{E^4}{f_\pi^4} < 30\pi \frac{c_s^4}{1 - c_s^2} \,. \tag{A.29}$$

Although taking $E = H$ is not well-defined, since we are far from flat space (where the unitarity calculations are performed) in that limit, we can still use this bound as a check on our interpretation of the apparent violation of de Sitter entropy bound. Throwing caution to the wind, we combine this inequalities using $E = H$ to find

$$\frac{8\pi^2}{3c_s} < \frac{H^4}{f_\pi^4} < 30\pi \frac{c_s^4}{1 - c_s^2} \,. \tag{A.30}$$

The only viable solutions to these inequalities occurs when

$$c_s > 0.68 \ . \tag{A.31}$$

We see that $c_s < 0.095$ clearly falls outside this region, so that it lies in the strongly coupled regime derived using this naive interpretation of the partial wave unitarity bound. The bound on $c_s$ from the de Sitter entropy is a clearly a weaker constraint, but has the advantage that it applies in the de Sitter background directly.

For $c_3 \neq 0$, the flat space perturbative unitarity bounds become more complicated, as the $s$-wave amplitude depends on $c_s$, $c_3$ and $c_4$ (see Eq. (A.26a)). A proper comparison of the de Sitter entropy and scattering based bounds would require the next $\left(4^{\text{th}}\right)$ order in the derivative expansion in the Fokker-Planck equation which is beyond the scope of this work.

# B    Solving the Fokker-Planck with Steepest Descents

In this appendix, we will present an alternate solution to the Fokker-Planck equation

$$\frac{\partial}{\partial \mathtt{t}} P(\zeta; \mathtt{t}) = \frac{\sigma^2}{2} \frac{\partial^2}{\partial \zeta^2} P(\zeta; \mathtt{t}) - \frac{\gamma_3}{3!} \frac{\partial^3}{\partial \zeta^3} P(\zeta; \mathtt{t}) \ . \tag{B.1}$$

The basic idea is to use the Fourier transform to simplify the derivatives with respect to $\zeta$. Specifically, if we define

$$P(\zeta; \mathtt{t}) = \int_{-\infty}^{\infty} \mathrm{d}k\, e^{-i\zeta k} P(k; \mathtt{t}) \tag{B.2}$$

then the Fokker-Planck equation becomes

$$\frac{\partial}{\partial \mathtt{t}} P(k; \mathtt{t}) = \left( -k^2 \frac{\sigma^2}{2} - ik^3 \frac{\gamma_3}{3!} \right) P(k; \mathtt{t}) \ . \tag{B.3}$$

This can be integrated to obtain

$$P(k; \mathtt{t}) = C \exp\left( -\frac{k^2 \sigma^2 \mathtt{t}}{2} \mathtt{t} - ik^3 \frac{\gamma_3}{3!} \mathtt{t} \right) \ . \tag{B.4}$$

To determine the probability distribution of $\zeta$, we take the inverse Fourier transform

$$P(\zeta; \mathtt{t}) = C \int_{-\infty}^{\infty} \mathrm{d}k \exp\left( -i\zeta k - k^2 \frac{\Delta_\zeta}{4\pi^2} \mathtt{t} - ik^3 \frac{\gamma_3}{3!} \mathtt{t} \right) \ . \tag{B.5}$$

For large $\mathtt{t}$, we might suspect we can calculate this integral using the method of steepest descents. Specifically, we can deform the $k$ contour in the complex plane so that is goes

through a point $k_\star$ such that

$$\frac{\mathrm{d}}{\mathrm{d}k}\left(-i\zeta k - k^2\frac{\sigma^2}{2}\mathtt{t} - ik^3\frac{\gamma_3}{3!}\mathtt{t}\right)\Bigg|_{k=k_\star} = 0 \ , \tag{B.6}$$

which occurs when

$$k_\star = i\frac{\sigma^2}{\gamma_3}\left(1 \pm \sqrt{1 + \frac{2\gamma_3\zeta}{\sigma^4\mathtt{t}}}\right) \ . \tag{B.7}$$

Noting $\gamma_3 \ll \Delta_\zeta$, we should take the $-$ solution, since it lies closer to the real axis. We will assume $8\pi^4\gamma_3\zeta/\mathtt{t} \ll 1$ so that

$$k_\star \simeq -i\frac{\zeta}{\sigma^2\mathtt{t}} + i\gamma_3\frac{\zeta^2}{2\sigma^6\mathtt{t}^2} \ . \tag{B.8}$$

The resulting probability distribution can be determined approximately by using $k = k_\star + \bar{k}$

$$P(\zeta;\mathtt{t}) \simeq C\exp\left(-\frac{\zeta^2}{2\sigma^2\mathtt{t}} + \frac{\gamma_3}{3!}\frac{\zeta^3}{\sigma^6\mathtt{t}^2}\right)\int_{-\infty}^{\infty}\mathrm{d}\bar{k}e^{-\mathtt{t}\Delta_\zeta\bar{k}^2/(2\pi^2)}$$

$$\simeq \frac{\tilde{C}}{\sqrt{\Delta_\zeta\mathtt{t}}}\exp\left(-\frac{\zeta^2}{2\sigma^2\mathtt{t}} + \frac{\gamma_3}{3!}\frac{\zeta^3}{\sigma^6\mathtt{t}^2}\right) \ , \tag{B.9}$$

where $C$ and $\tilde{C}$ are constants. Here we used the fact the integrand is analytic in the region enclosed by contours $k \in (-\infty,\infty)$ and $\bar{k} \in (-\infty,\infty)$ to obtain our final result.

This reproduces Eq. (3.14), which we derived from our exact solution. However, we also see that the method of steepest descents will become problematic when we take $2\gamma_3\zeta/(\sigma^4\mathtt{t}) \geq 1$. This corresponds to the same condition as above for the failure of the Stochastic framework to reliably calculate the tail of the distribution.

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
