# Peer review of "A Tail of Eternal Inflation"

_SciPost Physics_

## Round 2 · Referee Report · Anonymous (Referee 1) · 2022-8-2

Strengths

Well-written paper

Weaknesses

See report

Report

I was asked to referee this paper for another journal and I could not recommend it for publication. I was asked again to referee it for SciPost. I simply repeat my last report for the other journal focussing on the reason why I could not recommend it. (I would say the authors were unlucky to get the same referee twice and a second opinion may be more fair.)

I am afraid I disagree, as I said in my first report, with the authors: I do not think that it is surprising at all that the regime of eternal inflation is not under control for models with non-trivial inflaton self-interactions. This is actually well understood in the literature. For instance in JHEP 09 (2008) 036 0802.1067, their reference [49], there is a long discussion about this in Section 4. I quote: " There is an important qualitative difference between these non-minimal models and the simplest slow-roll inflation case: eternal inflation may lie outside the regime of validity of the effective field theory...” followed by a concrete explanations of why eternal inflation is not in the regime of the EFT when there are large derivative interactions. As I said in my previous report, this is also well understood in the primordial black hole literature. Notice that the problem has to do with loss of calculability at horizon crossing and it has nothing to do with what stochastic methods allow to resum, the cumulative effect of super-horizon fluctuations. I do not know why the authors had a different expectations.
The conclusions they reach is what was expected from the beginning, and I am not sure there is much to learn from the calculations. I am afraid I cannot recommend the paper for publication.

  • validity: low
  • significance: ok
  • originality: good
  • clarity: high
  • formatting: excellent
  • grammar: excellent

Author:  Timothy Cohen  on 2022-08-04  [id 2711]

(in reply to Report 1 on 2022-08-02)
Category:
reply to objection

Dear editor,

We would like to restate the referee’s response so that it is clear why it is being rejected. The referee stated that the paper should be rejected on the grounds that the result are obvious. We emphasize that there are a number of new technical results in this paper. Let us summarize these new results, that are being rejected on the grounds of being obvious:

  • Equation 2.10 - the most general form of Stochastic Inflation consistent with the nonlinear symmetries of $\zeta$. (The form is perhaps obvious in retrospect but has never been published, to our knowledge)

  • Equation 2.18 - the leading correction to Stochastic Inflation in single field inflation. This correction has never been published before, to our knowledge. Notice factors of 2 and $\pi$ which needed to be calculated carefully.

  • Equation 3.10 - the solution for the probability distribution of \zeta from Stochastic Inflation with the new term in 2.18

  • Equation 3.21 - the solution for the probability distribution of \zeta from Stochastic Inflation including all terms in equations 2.10.

  • Figure 2 - the region in parameter space, during inflation in our universe, where the solution in equation 3.10 gives eternal inflation.

Just to repeat, the author is rejecting all these new results on the grounds that they are obvious.

Finally, in the referee’s response, they state the following:

"Notice that the problem has to do with loss of calculability at horizon crossing”

In figure 2, the region inside the purple curve delineates where the EFT is weakly coupled at horizon crossing by a very conservative measure (it actually demands perturbative unitarity is satisfied at energies of 6H and not just H). The conclusion we draw from our new technical calculation, taken at face value, is that we would be in an eternally inflating phase, in our universe, for the range of parameters indicated by the blue region. We interpret this as a breakdown of the EFT approximation that were made in deriving our equations, as we do not think it is plausible that our patch of inflation is actually eternally inflating. We also provide a physical explanation for the breakdown associated with rare fluctuations. Even without such an explanation, the above quote from the referee is clearly in contradiction with figure 2. The referee is saying this is obvious yet is not acknowledging the principle reason we think it is surprising: by all known measures, the EFT appears to be weakly coupled at horizon crossing, and so the fact that calculability is lost is not obvious from the point of view of the EFT itself.

Anonymous on 2022-08-08  [id 2715]

(in reply to Timothy Cohen on 2022-08-04 [id 2711])

Dear Editor,

I would like to reply to what the authors wrote.
I want to explain once again why an inflationary theory with c_s << 1 is strongly coupled if you ask questions about \zeta ~ 1. If you compare the quadratic and cubic action, the ratio is \zeta/c_s^2. Now whether non-Gaussianities can be treated perturbatively or not depends on the size of \zeta, i.e. the question you are interested in. If you are interested in \zeta ~ 1 because you are asking a question about eternal inflation, or the formation of a primordial black hole, then the ratio above is large. (The plot the authors refer to is done looking at typical fluctuations, of order the rms, so it is not related to the regime \zeta ~1). This kind of non-perturbativity is at horizon crossing and it surely cannot be overcome by using stochastic method. This is the reason why I say there is no surprise in what the authors find. (It may well be that their results about the stochastic equation are relevant in some other context, but it seems to me they are of no use here.)

Author:  Timothy Cohen  on 2022-08-22  [id 2742]

(in reply to Anonymous Comment on 2022-08-08 [id 2715])

The referee claims that it is obvious that our calculation should break down when \zeta~1, based on estimates of the size of the respective terms in the action for \zeta. However, this neglects the fact that one of the primary features of Stochastic inflation is that it improves upon naive perturbation theory, exactly the same way that the renormalization group improves the naive Feynman diagram perturbation theory in quantum field theory.

Specifically, our explicit calculation is inconsistent with the referee's statement. This follows from our equation 3.14. One can plug \zeta \sim 1 into our solution for the probability distribution of zeta with H t >> 1. Then by counting powers of H t >> 1, one can see that the non-Gaussian term is parametrically smaller than the Gaussian term. (Using our equation 3.21, one can additionally see that this argument extends to theories with higher derivative terms.) This is why we can say with full confidence that there is no expected breakdown of Stochastic inflation when \zeta ~ 1.

We instead identify that even though Stochastic inflation improves the range of validity of expressions as a function of \zeta, there is nonetheless a regime where a breakdown of the theoretical predictions occur. The arguments presented in the previous literature fail to parametrically predict where our calculation breaks down. This is why we feel justified in suggesting that the result in this paper is not obvious based on the previous literature.

---

## Round 2 · Referee Report · Anonymous (Referee 2) · 2022-8-9

Report

This paper discusses the onset of slow-roll eternal inflation in the framework of the global description of eternally inflating spacetime (which is assumed implicitly). The authors argue that because of induced primordial non-Gaussianity, the region in which \Lambda < f_pi should be regarded as being outside the regime of the effective theory, despite the fact that cosmological correlators are under control by perturbation theory. In particular, the onset of eternal inflation is incalculable in this regime. Despite what the authors say, the result is very much expected in the sense that the effective theory breaks down when a low energy quantity controlling physical observables exceeds the cutoff scale. I would suggest that the authors revise the introduction and abstract to focus more on technical progress they made in analyzing the problem, rather than claiming/emphasizing the novelty and couterintuitiveness of the phenomenon (which I disagree with).

  • validity: -
  • significance: -
  • originality: -
  • clarity: -
  • formatting: -
  • grammar: -

Author:  Timothy Cohen  on 2022-08-22  [id 2743]

(in reply to Report 2 on 2022-08-09)

We disagree with the suggestion that we rewrite our paper. The reasons that justify our point of view are given in the latest public response to referee 1, so we will not repeat them here.

---

## Editorial Decision

resubmitted